# Diagnostics of Large-Panel Buildings—An Attempt to Reduce the Number of Destructive Tests

**DOI:** 10.3390/ma17010018

**Published:** 2023-12-20

**Authors:** Maciej Wardach, Janusz Ryszard Krentowski

**Affiliations:** Faculty of Civil Engineering and Environmental Sciences, Bialystok University of Technology, Wiejska 45E, 15-351 Bialystok, Poland; janusz@delta-av.com.pl

**Keywords:** condition assessment, prefabricated buildings, large-panel, destructive testing, NDT

## Abstract

Structural condition diagnostics provides the basis for decision making regarding the possibility of continued safe operation, necessary reinforcement, repair work, and in extreme cases, dismantling of the structure. The most reliable results concerning the condition and strength of materials are provided by destructive testing. However, these tests are very time-consuming, costly, and difficult to perform on in-service facilities. In addition, they involve the need to obtain the consent of the occupants of the premises and subsequent renovations. This article focuses on presenting an opportunity to reduce the number of destructive tests necessary to reliably assess the condition of large-panel structures, which constitute a significant housing stock in Europe. Based on tests carried out on a real building, the risk factors associated with obtaining reliable results by non-destructive methods were determined. Areas where destructive testing is necessary were identified. In addition, reference was made to standard recommendations and guidelines from a reputable research institution. Practical guidelines were formulated regarding the diagnostics of large-panel structures, resulting in a reduction in the number of destructive tests required.

## 1. Introduction

During the building design process, the architects work with the client to define the form of the building and adjust the layout of the rooms to suit the intended use. The designers then select structural solutions so that all the limit state conditions are met. This involves selecting materials with suitable strength properties, adopting and verifying optimum cross-sections of structural elements, and correctly designing their connections. In the case of prefabricated system buildings, the process of manufacturing and completing the elements begins once the design documentation has been approved. The last stage is the erection (assembly) phase, which continues until the building is handed over for use. Each of these phases of construction is subject to the risk of errors that may affect the durability of the completed building. Structures that have been in use for several decades are subject to a process of ageing of materials, as well as degradation resulting from improper use, environmental impact, or exposure to exceptional loads. Errors occurring during both the execution of the building and external factors occurring during operation create a risk of structural failure. In order to avoid undesirable phenomena associated with the failure of a structure or to identify ways of strengthening it should they occur, it is necessary to diagnose the actual condition of the building.

This article considers the research methodology for large-panel structures, which represent a significant proportion of Europe’s housing stock. Tens of millions of people live in these buildings, which makes ensuring safe and comfortable operation a matter of great social concern. During routine inspections, visual examinations are the most common practice, as they do not interfere with the structure of the building and are the least labour-intensive, but at the same time have the highest risk of error. If these inspections do not identify any signs of structural or component degradation, the inspections can be considered reliable and sufficient. However, if symptoms characteristic of structural degradation are present, or if an upgrade/reconstruction of the structure is planned, the condition assessment should be more precise. To this aim, tests should be carried out using methods that allow the strength parameters of the building materials to be determined. Destructive testing and excavation provide the most reliable data on the strength and condition of individual materials and structural elements. Performing destructive testing involves interfering with the structure of the building, which is highly undesirable in residential buildings due to the potential need for subsequent repairs to the premises. This factor poses problems for both the tenants and facility manager, who is burdened with covering the cost of the repair work.

Large-panel structures were erected in many countries, primarily in Central and Eastern Europe [1]. After many years of use, typically around 40–50 years, scientific and professional evaluation should focus on their potential for modernisation [2,3,4], analysis of occurring failures [5,6,7,8], and solutions aimed at enhancing their architectural qualities and energy efficiency by altering the facades [9,10]. All of these aspects are interconnected with the necessity of conducting a diagnostic assessment of the structure. Given the quantity of buildings, diagnostics should be carried out in an efficient manner, providing both rapid and reliable results.

This paper focuses on presenting the possibilities of reducing the number of destructive tests for assessing the condition of large-panel structures. Based on the conducted research, risk factors related to obtaining reliable results through non-destructive methods were identified, and areas where destructive tests are necessary were pinpointed. A selected large-panel structure located in Poland, constructed using the regional OWT system, was chosen for the study. The structure was in use for 33 years since 1978, decommissioned, and after another 10 years, a decision was made to dismantle it (Figure 1).

The analysed large-panel system was characterised by a structural module of 4.80 m × 5.40 m and a height of 2.70 m [11]. The 0.14 m thick prefabricated floor slabs were supported by walls, also 0.14 m thick, and on an external beam-wall. The edges of the wall slabs were dowelled at the prefabrication plant, and the spaces between adjacent walls were filled with concrete on site to form vertical concrete joints working in shear. Walls, floors, and beam-walls were connected during assembly by steel welded joints. The gable walls and beam-walls that formed the facade of the building were made of three layers. The facade layer was suspended on steel hangers to the bearing layer, and insulation in the form of wool or polystyrene was placed between these layers.

Performing structural diagnostics enables the selection of appropriate strengthening and repair methods. The tests are intended to provide both information about the strength parameters of the material being tested and the type and extent of deterioration. An example of the use of non-destructive testing to determine repair and rehabilitation strategies for a reinforced concrete structure is presented in [12]. Researchers have been working on methods of strengthening large-panel buildings for years. An extensive analysis in terms of the work required to increase the durability of these buildings and improve the comfort of their use is described in [13]. In [14], the author presents the possibilities of reinforcing vertical joints with a new steel connection system. The method of strengthening the connections of gable and beam-wall facade layers is presented in [15]. 

Widely practised methods of strengthening various types of concrete structures include reinforcement with composite tape [16,17] and repair mortar [18]. The repair materials used are extensively tested to determine their structural properties, their effect on the working of the structure, and their resistance to environmental influences. A very important issue is to subject the structure to testing after the strengthening work has been carried out. This is to determine the actual adhesion of the repair layer with the damaged structural elements. Correct adhesion is essential for the correct interaction of the old concrete with the newly added layer. An example of the use of the ultrasonic method to determine the adhesion of a repair layer is presented in [19]. To ensure proper adhesion to the concrete substrate, a suitable curing method is also required. In [20], the use of semi-destructive methods to test repair mortars applied using different curing methods is presented. The authors in [21] describe a repair material for which ohmic heat curing was used to obtain strength in a very low temperature environment. Interesting analyses are been presented in [22], wherein the effect of surface preparation on the adhesion of concrete for underwater structural repairs is been analysed. Researchers have also addressed the issue of self-healing cracks in concrete, as extensively discussed in [23]. Researchers have also been involved in analyses related to determining the influence of the external environment on the self-healing effect of concrete cracks [24].

Methods of strengthening structures that are located in seismically active zones have also been analysed. In [25], studies related to the placement of inter-storey seismic isolation in a wall-frame building are presented to determine the best location of the isolation at the height of the structure, which can significantly contribute to the durability of seismically exposed structures. Analyses related to the design of earthquake-resistant structures are also described in [26], wherein a seismic design concept is presented in which a building is divided into several segments, connected to one another by additional vibration isolation systems.

As presented in [27,28,29], large-panel buildings are characterised by numerous errors arising during production and installation, as well as degradation resulting from use often extending over 50 years. A number of papers can be found in the literature in which studies of large-panel buildings are presented [30,31,32,33,34]. Despite the development of non-destructive testing apparatus and its increased availability, there are still doubts about the effectiveness and validity of its use in practice. The main problem is the lack of testing methodologies. These should concern specific types of structures, characterised by their typical structural solutions and defects, and take into account the possibility of using various testing methods, including non-destructive ones. This article is a continuation of the analyses related to the improvement of testing methodology for large-panel structures and the implementation of modern non-destructive testing methods in engineering practice. The authors described their earlier analyses in detail [35,36,37,38,39], wherein they presented a wide range of tests using different methods. The research results presented in this article are novel and have not been taken from other sources. The development of technology allows for a change in the techniques/methods used in the assessment of structures, which can speed up diagnostic processes and reduce the number of destructive tests. The authors have attempted to answer the research question of whether a large number of destructive tests are necessary to diagnose structures and in which areas destructive testing is necessary to carry out a reliable assessment of the building’s condition.

The body of knowledge on non-destructive methods is very broad, as exemplified by the numerous papers on the subject [40,41,42,43,44]. This article is a continuation of the extensive literature on non-destructive testing, but focuses strictly on the specificity of large-panel buildings. Thus, it tries to respond to the need to develop dedicated and precise research methodologies for system structures. The manuscript concentrates on the aspect of using non-destructive methods that can improve the diagnostic processes of thousands of large-panel structures. With regard to the application of the testing methods used, reference was made to standard recommendations and guidelines from a reputable research institution. Based on the performed research, practical guidelines were formulated with a view to reduce the number of destructive tests and increase the efficiency of conducted structural condition assessments. The unique opportunity to carry out such extensive tests on a large-panel object provides a unique opportunity to improve diagnostic techniques.

## 2. Structural Diagnostics—Own Research

In large-panel construction, two main structural materials were traditionally used: concrete and steel. Load-bearing elements, such as wall panels, floor slabs, beams, beam-walls, stairs, and foundations, were made of reinforced concrete with steel rebars. Steel was also used in constructing joints connecting the prefabricated elements. The condition of structural elements is influenced by the degree of degradation in both concrete and steel. Due to the intended demolition of the building, the authors were not restricted in the number and scope of conducted destructive tests. Hence, dozens (about 80 units) of core drillings and numerous excavations were carried out, along with cutting reinforcement bars and connection plates for subsequent laboratory strength tests (Figure 2). The unique opportunity to obtain such a large number of samples enabled the discovery of empirical relationships between results obtained from various methods.

### 2.1. Methods of Assesing the Quality and Strength of Concrete

Concrete, as a material consisting of aggregate, cement, water, and various admixtures and additives, is characterised by different physical and strength properties. The properties of concrete depend on both the composition and environmental conditions in which it is formed/bonded, as well as the method of placement and compaction itself. In addition, concrete elements are exposed to environmental influences that can significantly affect their strength properties, as well as their ability to protect reinforcing steel from corrosion.

Many test methods dedicated to the assessment of concrete quality have been described in the literature [45,46,47,48,49,50]. Many of the modern techniques are usually applied under laboratory conditions [51,52,53]. In construction practice, a visual method is first used to assess the condition of reinforced concrete structures, allowing cracks and scratches to be observed, which can provide a basis for deciding whether more thorough testing is necessary. Subsequently, sclerometric, ultrasonic, and destructive testing methods are used to assess the condition of concrete and reinforced concrete elements. As these three methods are the most commonly used, the authors have selected them for analysis with the aim of reducing the number of destructive tests in construction practice.

By using samples obtained by core drilling, we can acquire many parameters, such as compressive strength, tensile strength, Young’s modulus, water resistance, information on the extent of carbonation, the composition of the concrete mix, and many others. The basic question is which parameters are necessary to assess the condition of a building that has been in service for several decades. Many times such buildings do not show any symptoms indicating that their durability is at risk. It is therefore reasonable to determine the purpose of the diagnosis to be carried out. A thorough analysis, including the composition of the mixture, may be necessary in exceptional situations, such as a building catastrophe, to verify that the structure was designed and constructed in accordance with the applicable standards. For regular condition monitoring, or to perform an expert assessment prior to retrofitting works, basic strength parameters and identification of degradation processes may be sufficient to determine. Large-panel structures often maintain a technical state that does not pose a threat of failure, partly due to latent safety reserves, so analyses of the condition of reinforced concrete elements can be simplified to some extent in many cases.

In residential buildings, floor slabs are covered with floor layers, while walls are usually plastered and painted or wallpapered. For non-destructive testing, it is often required to remove unnecessary layers from the walls, whereas in the case of the floor slabs, it is reasonable to test from underneath, eliminating the need to remove the floor layers. When collecting samples for destructive testing, drilling rigs are used, which are usually water-cooled. Water degrades large areas of the finish layers, including those of the flats on the storey below, and it is additionally necessary to mask the entire newly created cavity. The common opinion perceiving destructive surveys as the most problematic and labour-intensive, both at the time they are carried out and in order to bring the flats to a condition equivalent to before they were started, is justified.

In order to determine the strength of the concrete, the authors carried out numerous tests using sclerometric (Figure 3a) and ultrasonic methods (Figure 3b), followed by core drilling samples (Figure 3c) at the same locations for destructive strength tests. In order to check the suitability of the individual testing techniques, the results obtained using destructive and non-destructive methods were correlated according to the procedures described in the standard [54] and its previous version [55], which described some aspects of the testing in more detail. The analyses were also based on recommendations (instructions) of the Building Research Institute (in Polish: ‘Instytut Techniki Budowalnej’—‘ITB’) [56,57,58], a leading research institution in Poland, which are very often practised by engineers. European standards and the guidelines of national research institutes are the most reliable source of knowledge for building surveyors, so the authors focused on assessing the suitability of their recommendations for the diagnosis of large-panel structures. 

The regression curves were determined based on the correlations obtained between samples collected from core drillings made on the 4 interior walls of the 5th floor of the building under analysis. Three core drillings were made on each wall (with a diameter of *d* = 10 cm, which were cut to obtain *l/d* = 1). Before core drilling, the number of rebounds was read 10 times at each measuring point to determine the median. At the same locations, the ultrasonic wave velocity was measured using the direct method, i.e., by placing transducers on 2 sides of the walls. 

#### 2.1.1. Concrete Compressive Strength Testing by Sclerometric Method

The first of the methods analysed for testing the compressive strength of concrete was the sclerometric method. This method is distinguished by its low effort, simplicity, and availability of equipment. This makes it an attractive and widely used technique. The results obtained using the testing machine and sclerometer are presented in Table 1, which also includes the differences between the obtained values (Δf). Both the most common regression curves proposed by the standard [55] (denoted by f_R,EN_) and the ITB instructions [56,57] (f_R,ITB_) were used for the analyses. Using the data from the destructive tests (f_is_), empirical relationships were determined and corrected curves were determined—both with the approximate method (f_R,EN,scaled_) according to [55], and with the exact method, i.e., the least squares method (f_R,accurate_). In addition, a regression curve with strength reduction factors due to the age of the concrete was determined according to the recommendations of the ITB instructions [56,57] (f_R,ITB,reduced_). All curves are presented in Figure 4. 

The obtained results showed that, in the case of concrete made more than 40 years ago, the compressive strength calculated on the basis of the regression curves proposed in [55] and refs. [56,57] was significantly higher (by about 30%) than the actual values. This fact can be explained by the phenomenon of concrete carbonatisation (overestimating the reflection number), which, when measured with phenolphthalein solution, ranged from 1 to 3.7 cm deep in the tested walls. The smallest differences between the obtained results were represented by the equation of the curve determined using the exact method (difference of an average of 1%). The equation determined using the approximate method produced results that deviated from the actual results by an average of 3%. The regression curve recommended by the ITB instructions, which takes into account the reduction in strength of the concrete due to its age, underestimated the results by 14% on average.

For further analyses, curves described by two equations were selected, i.e., the regression curves f_R,accurate_ and f_R,ITB,reduced_. The choice was based on the fact that the results obtained with the curve obtained using the exact method were the most precise, while the second equation underestimated the strength of the concrete, but this curve did not need to be scaled, which is a huge convenience for the diagnostician in practice. In addition, obtaining lower results than the actual values is safer from the point of view of the structural surveyor because the difference creates a hidden safety reserve. However, the caveat is that if the analysis carried out on the basis of the results obtained in this way indicates that the limit states are not met, it is imperative that the parameters should be confirmed by a destructive method before a reinforcement decision is taken. This is to avoid unnecessary expense on works to increase the strength of the structure.

Large-panel buildings were assembled from prefabricated elements which, for a particular structure, were most often made in a single factory using the same concrete mix. Hence, it can be assumed that the concrete quality and strength of the elements located on different storeys should be similar. In order to verify that the selected curves were applicable to the determination of the compressive strength of the other building elements, a comparison was made between the results obtained from 12 core drillings (f_is_) taken from the walls and floors of storeys 1, 3, 7, and 10. The results are presented in Table 2 and Figure 5. The difference between the obtained strength values is denoted as Δf.

Based on the obtained results, it can be concluded that the compressive strength of the concrete calculated using the equation determined from the empirical relationships established for the walls of the 5th floor was very close to the actual compressive strengths of the concrete of the other prefabricated elements located on storeys 1, 3, 7, and 10. The maximum overestimation of the concrete strength was 12%, while the underestimation was 10%. When the regression curve recommended by the Building Research Institute (ITB) was applied (i.e., strength reduced due to the age of the concrete), the concrete strength was slightly (by 4%) overestimated only for one sample. In the remaining cases, the maximum underestimation was 24%. The above analysis confirmed that the concrete used throughout the building was characterised by similar strengths, irrespective of the storey or type of structural element. The above relationship may be a typical feature of buildings constructed using large-panel technology.

The current standard [54] indicates that, for the purpose of assessing an existing structure, an estimate of the compressive strength should be based on at least 8 or 12 valid measurement values, depending on the diameter of the core. On the other hand, for a small measurement site that contains 1 to 3 structural elements, the total volume of which does not exceed 10 m^3^, a minimum of 3 drillings should be made. The standard also indicates that the calibration of results obtained using indirect methods (non-destructive methods) should include at least 10 pairs of results. It is also permissible to use indirect methods without calibration to locate sites of questionable quality, after which a minimum of 3 core drillings should be carried out. This method is limited to measurement locations where there is no doubt about the compressive strength of the concrete and the total volume of the elements does not exceed 30 m^3^.

The authors, based on their professional experience, their knowledge of large-panel buildings (especially those constructed in OWT system), and the studies carried out, recommend that the concrete throughout the structure should be assumed to represent the same compressive strength class. This makes it possible to adopt, as shown in the flowchart proposed in the standard [54], a procedure that allows the structure to be treated as a single measuring site. It is therefore suggested to take core drilling samples from different structural elements located on different storeys, instead of taking several cores from one/multiple elements located on one storey. In this way, the structural surveyor can obtain information about the condition of the concrete in the entire structure without performing a large number of destructive tests. In a further step, empirical relationships should be determined with the results obtained using the sclerometric method. Subsequent elements are suggested to be tested exclusively using the sclerometric method to estimate the concrete compressive strength. In the case of locating areas with an extremely different number of rebounds, a core drilling should be carried out in this location. In situations where drilling is not possible, it is not suggested to use the curve recommended by the standard [55] without calibration. Instead, it is recommended to use the curve proposed by ITB [56,57], taking into account the reduction in compressive strength of the concrete due to age. The results may deviate considerably from the actual results, but it should be assumed that it is better to estimate the result than to not to have it at all in such a case.

#### 2.1.2. Concrete Compressive Strength Testing by Ultrasonic Method

Another method to be analysed for estimating the compressive strength of concrete was the ultrasonic method. There is no direct correlation between strength and ultrasonic wave speed. The relationship should be established for a specific concrete mix recipe. The results are also influenced by factors such as the moisture content and temperature of the concrete, the location of the reinforcement, as well as cracks and voids. The moisture content of the concrete and its temperature affect the velocity of wave propagation. Wave propagation can be faster in wet concrete. The temperature of the concrete under normal conditions does not significantly affect the velocity and attenuation of waves, except when the concrete is subjected to freezing. All elements were tested a few days apart, during the summer season, where the outside temperature did not exceed 25 °C in the day and did not drop below 10 °C at night. There was also no precipitation at that time. According to the test equipment producer’s instructions [59], no correction factors were implemented for concrete at temperatures between 10 °C and 30 °C, regardless of the degree of moisture in the material. As the measuring conditions were the same throughout the test period, and following the producer’s instructions, the influence of temperature and moisture was neglected. Another factor is the influence of the location of the reinforcement. The velocity of the ultrasonic wave through rebar is much higher than in concrete. Measurement locations through which the reinforcement passes should therefore be avoided. For the purposes of this study, scanning of the tested elements using the electromagnetic method was carried out in order to locate the rebars. Measurement locations without reinforcement were then determined. In this way, the influence of the reinforcement on the results was avoided. The last factor mentioned above is the presence of cracks and voids. Cracks and voids with a length greater than the diameter of the transducer or the wavelength in the path of the ultrasonic pulse present an obstacle that causes the wave energy to dissipate. This causes the wave transit time to be extended. In the tested elements, the ultrasonic wave velocity and rebound number qualified the concrete mix as homogeneous and of relatively good quality. In addition, visual examinations were carried out after the core drilling, confirming the good quality of the concrete, with no internal cracks or voids at the designated measuring points.

Testing the speed of the wave propagation is recommended for locating areas of varying quality. 

Both the standard [55] and ITB instructions [58] provide the equations of the basic regression curves, the shapes of which are shown in Figure 6. On the basis of the obtained results of our own tests (Table 3) and using the producer’s software, the empirical relationships and equation of the curve were determined (f_V,accurate_), which are also shown in Figure 6. The compressive strength of concrete determined using the destructive method is denoted by the symbol f_is_.

As can be seen in Figure 6, the shapes of the curves are quite different, so estimating the compressive strength of concrete without determining empirical relationships is unreliable. Therefore, standard curves should not be adopted without scaling, and if this is not possible, the use of the ultrasonic method should be limited to locating areas of poorer quality. 

Similar to the sclerometric method, the authors verified that the determined curve was applicable to the reliable determination of the strength of other structural elements. The relationships were verified for the same measurement points as in the analyses of the sclerometric method. The results are presented in Table 4 and Figure 7.

The maximum difference in the compressive strength of the concrete was 9%, demonstrating that the selected regression curve was applicable for reliable estimation of concrete strength in the whole tested structure, without differentiation by storey and type of prefabricated elements. This demonstrated the use of a concrete mix of the same, or very similar, composition for the prefabricated elements, as well as the same standards of workmanship of the elements at the prefabrication plant.

#### 2.1.3. Testing the Quality/Homogeneity of Concrete Using Sclerometric and Ultrasonic Methods

Both the sclerometric and ultrasonic methods are commonly used to estimate the quality/homogeneity of a concrete mix. The guidelines contained in ITB [56] enable the homogeneity of concrete to be assessed on the basis of the calculated coefficient of variation of concrete compressive strength (*v_f_*), which is determined using the sclerometric method. By contrast, ref. [60] presents a classification of concrete quality based on the velocity of longitudinal wave propagation in the element. The authors decided to compare the results of the two methods using their own tests carried out on 5 internal walls, for which the mean velocity from 10 longitudinal wave readings and the median from 10 reflection number readings were obtained. The results are presented in Table 5. The calculated coefficient of variation of concrete compressive strength was calculated according to the guidelines in [56].

Based on the presented results, it can be concluded that the tested concrete was of good quality. The range of longitudinal wave velocity values contained in [60] was quite wide, resulting in all of the elements tested being assigned to one quality category. When the sclerometric method was used, the homogeneity of the concrete was classified as medium, good, or very good. In the case of assessing the homogeneity as ‘medium’, a value of 14% was the lower limit of this category; hence, taking into account the other results, the concrete could be considered homogeneous. Analysing the above table, it was concluded that both methods converged and could be successfully used to assess the quality and homogeneity of concrete. However, it is recommended to compare the results obtained using the two methods. Using only surface methods (e.g., the sclerometric method) runs the risk of not identifying some areas of poorer quality, which can be successfully detected by ultrasonic methods, as described in [36].

### 2.2. Methods of Assessing the Condition and Strength of Steel

In large-panel buildings, steel is present in structural elements in the form of reinforcement bars and welded joints of prefabricated elements. Degradation of this material is difficult to identify due to it being covered in concrete. The reinforcing bars are wrapped in the concrete mixture forming an integral whole, while the steel connections are hidden under other reinforced concrete elements, i.e., walls, floors, and beam-walls.

Corrosion of the reinforcement leads both to a reduction in the load-bearing capacity of the reinforced concrete elements (e.g., through a reduction in rebar cross-section, loss of adhesion to the concrete, or loss of ductility). In order to assess the quality of the reinforcement works, electromagnetic methods are most often used or excavations are carried out. Modern measuring equipment makes it possible to determine the diameters and spacing of the reinforcement without the need to remove finish layers. Carrying out a series of tests in a building which has been used for its intended purpose for several decades and also exposed to a more aggressive environment for a short period of time (as a result of being taken out of service and not being heated) can provide a reference for an approximate assessment of the condition of the reinforcement of the load-bearing elements in other large-panel buildings. In order to determine the strength parameters and quality of the reinforcement works, a dozen excavations were carried out, several dozen rebars (more than 100) were cut for destructive testing, and tests were carried out using the electromagnetic method. Comparison of the test results obtained using the electromagnetic method and destructive methods made it possible to determine the accuracy of the non-destructive method in relation to the actual state and to locate sensitive areas where special care should be taken when taking measurements and interpreting the data. Precise determination of the strength parameters of the reinforcing bars using destructive testing allowed visualisation of the expected state of rebar degradation in a large-panel building more than 40 years after installation. Strength tests were also carried out on the steel connections of the prefabricated elements, which are responsible for the proper redistribution of internal forces in the structure.

#### 2.2.1. Testing the Quality of Reinforcement Works and Strength of Bars

The authors first carried out electromagnetic measurements (Figure 8a) of the walls, floors, and beam-walls to determine the location of rebars and quality of the reinforcement works. Several of these locations were marked for excavation and sampling for laboratory testing (Figure 8b,c). The diameter measurement made with the apparatus, according to the manufacturer’s specifications [59], can be subject to an error of ±2 mm. It should be noted that in the case of ribbed bars, a distinction was made between the basic diameter of the core and height of the ribs, and the declared diameter was a dimension defined in product-specific standards. The measuring device did not distinguish between smooth and ribbed rebars, hence it was necessary to be careful in interpreting the results. The authors measured the actual diameters of the bars after excavation using a calliper and micrometer and then determined their declared diameter, taking into account the dimensional tolerances given in refs. [61,62], i.e., the local standards in use at the time when the structure was erected. Based on the ribbing, the steel grades were identified, which are presented in Table 6. The Φ8 diameter bars were smooth reinforcement, while the bars of the other diameters were ribbed. Examples of the values obtained were compared with the electromagnetic test results, as shown in Table 6.

The obtained results supported the conclusion that the deviations in the measurement of diameter using the electromagnetic method were within the range declared by the device manufacturer. In addition, the most frequent reading for each diameter type indicated the design value, which made it easier to interpret the results. However, it should be noted that the identification of the correct diameter should also be based on the diagnostician’s experience and knowledge of design solutions. For example, bars with a nominal (design) diameter of Φ15 were not used in Poland. If the readings at a given location were Φ14 or Φ15, it should be assumed that Φ14 bars were used in the construction. It is worth mentioning that all of the measured bar diameters were within the upper limits of the dimensional tolerances, which is beneficial for the load-bearing capacity of reinforced concrete elements.

Dozens of rebars (10 to 20 for each diameter type) were tested in a machine to determine their yield (f_y_) and tensile strength (f_u_) (Figure 9). The specimens were machine-cut to obtain a full cross-sectional area (without ribbing) in order to precisely determine the strength parameters tested. The results are presented in Table 7.

The actual yield strength values were between 12 and 65% higher than the minimum values for the identified steel grades specified in the standard [63] in use at the time when the facility was designed. None of the individual specimen results indicated that the yield strength value was too low—all rebars met the standard strength requirements.

#### 2.2.2. Concrete Resistivity Testing—Estimating the Risk of Reinforcement Corrosion Processes

At the same measuring points, a resistivity test of the concrete was also carried out prior to the excavations. This test is especially useful for assessing the resistance of concrete to the ingress of chloride ions, which cause corrosion of the steel. The chlorides that initiate steel degradation processes come primarily from de-icing salts to which car parks and underground garages are exposed, as well as chlorides from seawater and air. In the case of large-panel structures, the danger of chloride aggression occurs on coastal shores in the form of wind-borne aerosol and in areas of large industrial districts. The resistivity test method is also used to estimate the corrosion rate of already-depassed steel. A classification of the corrosion intensity, divided into chloride-induced and general corrosion depending on the resistivity of the concrete, is presented in [64]. The results obtained using Wenner probe measurement are presented in Figure 10 and the test equipment is shown in Figure 11a.

The obtained results indicated that, in a facility not located by the sea and not in the zone of a large industrial district, the probability of chloride corrosion was assessed as very low. The results (>500 Ωm) also indicated that the conditions in the concrete cover were not conducive to rapid corrosion of the reinforcing steel. The absence of corrosion signs of the rebars was confirmed by visual inspection of the cut specimens (Figure 11b). However, corrosion of the reinforcement was identified in the area of the window joinery (Figure 11c), which occurred as a result of the bars being exposed by inadequate dismantling of the joinery, and in the facade hangers of the textured (facade) layers, the concrete resistivity of which was not tested due to lack of access.

#### 2.2.3. Testing the Strength of Steel Connections

The most difficult places to carry out condition assessments are the steel connections of prefabricated elements. As described in [35], in order to reduce nuisance to residents, it is recommended that joints should be tested in stairwell areas. By assessing the condition of several such connections, including the conformity of their workmanship with catalogue solutions, it is possible to determine with what care the connections were made and whether degradation processes (e.g., in the form of corrosion) are occurring. If the connections in a stairwell are identified as being in poor condition, it is highly likely that the connections throughout the building are in a similar state. When exposing a steel joint, it is highly desirable to determine the condition without cutting it out, which would result in a loss of load-bearing capacity. To do this, the surfaces should be brushed out using power tools, and then non-destructive testing should be carried out. Ultrasonic hardness tests are helpful for estimating the strength of steel, while plate thickness can be successfully read using ultrasonic thickness gauges. The number of destructive tests should be limited to as few as possible. In order to check whether the results for the tensile strength of the steel obtained by correlation with its hardness were reliable, the authors cut several plates of the connections (Figure 12), which were tested in a testing machine. From the results, the differences (Δf) between the values obtained using the destructive (f_u,is_) and non-destructive methods (f_u_) were determined, as shown in Table 8.

The obtained results allowed the tested steel to be classified as St3SX with a minimum tensile strength f_u_ = 360 MPa [65], which was in accordance with the design assumptions. This grade was most similar to the currently used equivalent with the symbol S235. The tensile strength values obtained using the non-destructive method and in the testing machine were very similar, and the maximum difference in strength was less than 6%. The clear correlation between the steel’s hardness and its strength made it possible to reduce destructive testing to the necessary minimum. The authors recommend that only part of the section (e.g., a section of angle) should be cut out in order to minimise the loss of load-bearing capacity due to damage of the joint. Therefore, it is suggested to avoid cutting whole flat bars connecting precast elements. In the case of similar results for the hardness of steel on different types of joint, the minimum number of destructive tests is recommended to determine the average tensile strength, i.e., 3 specimens. In the event of large discrepancies, areas of lower hardness should be identified using a non-destructive method, after which a sample should be taken for strength testing.

## 3. Discussion—Practical Aspects of Diagnostics

Using specialised measuring equipment to determine the actual strength characteristics, followed by FEA calculation software, it is possible to determine when a component under analysis will be destroyed with a high degree of accuracy. Non-destructive testing is always subject to the risk of error, whether this is caused by failure to take into account the influence of any of the factors affecting the measuring equipment, incorrect use of the equipment, or misinterpretation of the results. Destructive testing provides the most reliable results and a basis for confirming the reliability of data obtained using other methods. Extensive tests carried out on the analysed large-panel object enabled the identification of differences between the results obtained using destructive methods and the most commonly used non-destructive methods.

The obtained results showed a convergence of the concrete compressive strengths of the different types of precast elements and their good quality. The prefabricated elements were made in factories in which care was taken to maintain the technological regime, which may have had a positive effect on their quality and durability. In the authors’ opinion, for the diagnosis of large-panel structures where visual inspection does not indicate disturbing degradation processes and the buildings are not located in highly chemically contaminated zones, the number of destructive tests can be reduced to the necessary minimum. Standard recommendations for obtaining the required number of pairs of results obtained using destructive and non-destructive methods may not be possible in some cases. In such situations, building surveyors should base their decisions on their own experience and the recommendations given in articles such as this paper. In the authors’ opinion, with the knowledge of the specifics of the construction of a given building (the knowledge that the elements may be made of the same class of concrete) and the convergence of results obtained using non-destructive methods indicating good concrete quality, the number of drillings can be limited to a few per whole object. The number of drillings should be adequate to the number of storeys and the sampling locations should be on different structural elements and storeys. In this way, the results obtained using non-destructive methods can be confirmed with a minimum workload. It should be noted that when carrying out non-destructive testing, both the equipment producers’ recommendations and applicable standards should be strictly respected. Otherwise, the results obtained may be falsified, which may lead to wrong decisions being taken regarding the condition of the structure.

Tests related to the determination of the quality of the reinforcement works can be successfully carried out using non-destructive methods. A comparison of the results in terms of rebar spacing and diameter showed no significant differences between the readings obtained using the electromagnetic method and the measurements from the excavations. Resistivity measurements have also been shown to be helpful in determining whether the environment is conducive to the formation of corrosion processes. However, it is not possible to determine the strength parameters of the reinforcement using non-destructive methods. Tests carried out on dozens of rebars in the testing machine made it possible to conclude unequivocally that the reinforcing steel in the investigated object had not been subjected to degradation related to the long period of exploitation (ageing processes) and the influence of factors in the form of moisture from the premises and external environment, temperature, and atmospheric pollutants. It is noteworthy that the degradation processes of the reinforcing steel did not occur despite the fact that the building was out of use for a decade, which was associated with non-heating of the object in winter conditions and acts of vandalism in the form of broken glass in the windows. The authors recommend performing a minimum number of openings in selected structural elements and taking a few rebars in order to obtain certainty about the absence of corrosion processes and the preservation of the strength parameters of the steel. It is reasonable to excavate the gable walls, which are also bathroom walls, and basement walls. These are the elements most exposed to moisture from both inside and outside the building. Rebars passing through retrieved reinforced concrete cores can also be used for testing, which in some cases can eliminate the need for excavations.

Despite the absence of any mention in the literature regarding the identification of symptoms indicative of advanced degradation processes in large-panel structures in service, regular assessment of their technical condition is very important. In addition to the degradation processes associated with material ageing and inappropriate use, climate change and the strong industrialisation of urban regions are a threat. The changes taking place in the environment result in atmospheric factors (snow, wind, rain, and temperature) of a different intensity than was expected when the facilities were designed. Industrialisation, meanwhile, causes air and water pollution, which can contribute to the degradation of construction materials. These factors make it necessary to monitor the conditions of structures to ensure their safe operation.

## 4. Conclusions

In conclusion, on the basis of professional experience, previous research, and the analyses presented in this article, conclusions have been made regarding the diagnosis of large-panel structures aimed at reducing the number of destructive tests.

A number of research limitations were identified in the area of testing systemic prefabricated objects, resulting in recommendations to improve the diagnostic process:
Problems associated with determining the location and number of survey points. The authors recommend starting the survey by performing a visual inspection to assess the condition of the object and identify degraded areas. Then, carry out non-destructive tests to provide a basis for selecting sites for taking cores in areas of questionable quality.Lack of access to archival documentation of the investigated building. This makes it difficult to identify the original structural assumptions and solutions. In the absence of such documentation, it is recommended to make an initial assumption that all main structural elements were made of the same class of concrete.Uncertainty of results obtained using non-destructive methods. The results obtained using these methods are influenced by many factors, including moisture content, temperature, carbonisation processes, reinforcement, cracking and voids. In order to minimise the risk of obtaining an incorrect result, it is recommended to carry out comparative tests with two non-destructive methods. The results need to be confirmed by data obtained from destructive testing. These data should form the basis for finding empirical relationships between the destructive method and non-destructive methods: ultrasonic and sclerometric. When estimating the strength of concrete in subsequent structural elements already without confirming the results with cores, measurement by two non-destructive methods increases confidence in the data obtained. It also makes it possible to locate questionable areas in the event of differing values being obtained at the measuring point under examination, which provides an indication of the need for further destructive testing.Lack of possibility for destructive testing. In the absence of core drilling, estimating the compressive strength of concrete based solely on non-destructive methods is subject to a high risk of error. However, based on professional experience, the authors have sometimes found themselves in situations in which they have been forced to suggest only the results obtained using non-destructive methods. At that time, the guiding principle was that it was better to have estimates as a basis for any analysis than to be based only on visual assessment.

The analyses carried out also allowed conclusions to be drawn regarding the application of various test methods for assessing the condition of large-panel structures and the formulation of guidelines to reduce the workload involved in carrying out the tests:The assessment of the compressive strength of concrete, when core drilling is not possible, should be based on the use of the sclerometric method and regression curves that take into account the age of the concrete. The ultrasonic method is then proposed to assess the quality of the concrete and locate areas of questionable quality.Determination of the cover thickness and spacing of the reinforcement can be successfully carried out using the electromagnetic method. However, the sensitive data are the rebar diameters, as it is necessary to take into account the measuring accuracy of the instrument and type of rebar (ribbed/smooth).In most cases, it may not be necessary to assess the tensile strength of the reinforcing steel. However, if it is required to assess the strength parameters of the steel, it is recommended to identify the area where the rebars run and then to drill a core. This core can be used in two ways: first to determine the strength of the concrete and then, after breaking, to obtain a sample of the reinforcement. The rebar can be visually examined to assess corrosion processes and then be tested in a machine. According to the authors’ numerous experiences, the effect of the reinforcement placed transverse to the drill axis on the compressive strength of the concrete can be neglected in most cases. This reduces the need for performing additional excavations.A concrete resistivity test can be helpful to assess whether the concrete cover is conducive to corrosion processes of the reinforcing steel. This test is recommended to be carried out in the area of planned excavations or drillings in order to be able to confirm the results.The tensile strength of steel connection plates can be successfully determined by correlation with its hardness, using ultrasonic hardness testers. Destructive tests can be reduced to the minimum number necessary to confirm the parameters obtained using non-destructive methods.

Improving the methodology for assessing the condition of large-panel buildings contributes to reducing the labour intensity and, at the same time, the costs associated with carrying out surveys. This is particularly important for property managers who, due to the age of the buildings, will increasingly be obliged to carry out repairs, modernisations, or strengthening of the structures and thus perform condition assessments. 

The authors also identified the following research gaps related to non-destructive testing of large-panel structures:Lack of equipment to identify degradation of steel joints without performing excavations.Lack of research methodologies related to the assessment of the degradation status of facade layer connections in gable walls and beam-walls in modernised buildings.

A potential technique that can contribute to solving these research problems is computed tomography, the use of which will be an area of the authors’ future research.

## Figures and Tables

**Figure 1 materials-17-00018-f001:**
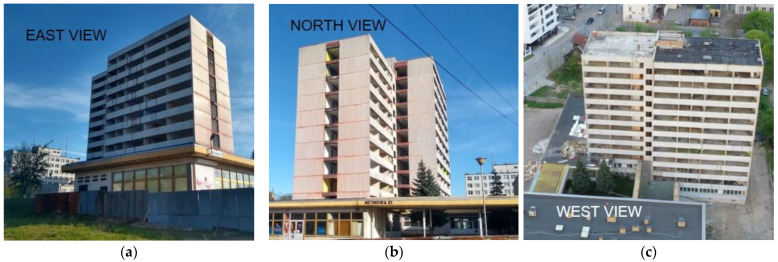
Examined large-panel building: (**a**) east side; (**b**) north side; (**c**) west side.

**Figure 2 materials-17-00018-f002:**
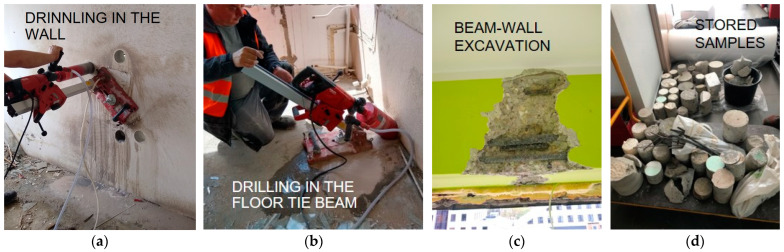
Destructive testing: (**a**) Core sampling from the wall panel; (**b**) core sampling from the floor tie beam; (**c**) excavation in the beam-wall; (**d**) image of some of the samples taken.

**Figure 3 materials-17-00018-f003:**
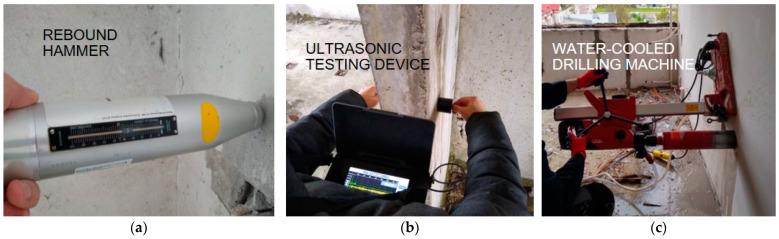
Concrete testing: (**a**) sclerometric method; (**b**) ultrasonic method; (**c**) drilling core sampling.

**Figure 4 materials-17-00018-f004:**
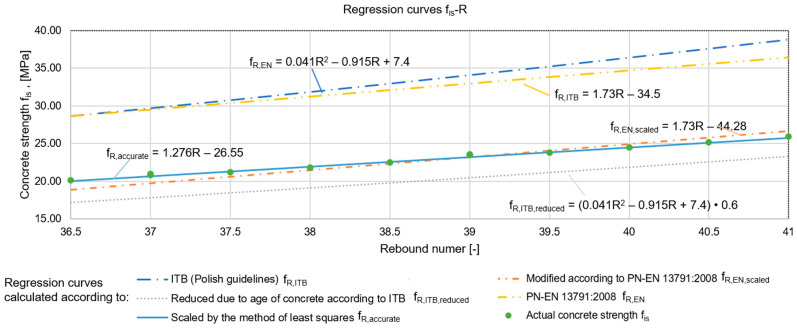
Regression curves for estimating the compressive strength of concrete using the scleroemtric method and results of destructive strength tests.

**Figure 5 materials-17-00018-f005:**
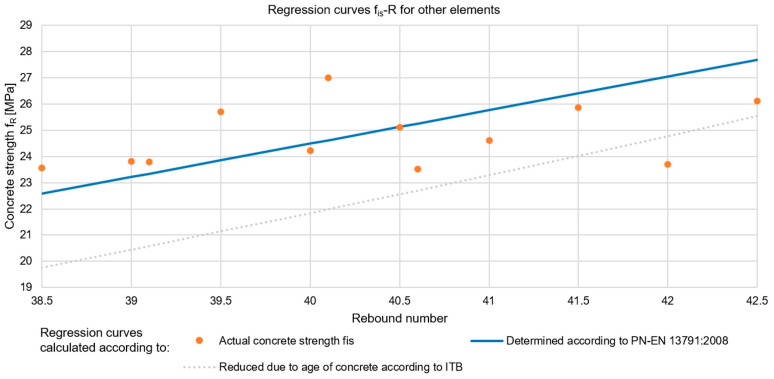
Regression curves and results of destructive strength tests for various structural elements located on storeys 1, 3, 7, and 10 (using the sclerometric method).

**Figure 6 materials-17-00018-f006:**
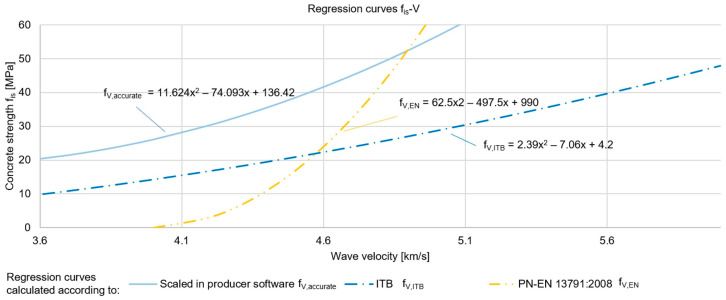
Regression curves for estimating the compressive strength of concrete using the ultrasonic method.

**Figure 7 materials-17-00018-f007:**
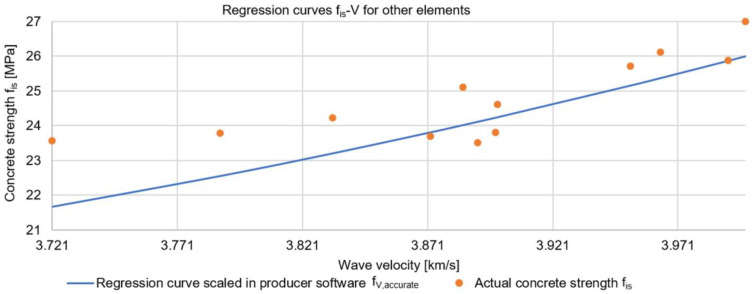
Regression curve and results of destructive strength tests for various structural elements located on storeys 1, 3, 7, and 10 (using the ultrasonic method).

**Figure 8 materials-17-00018-f008:**
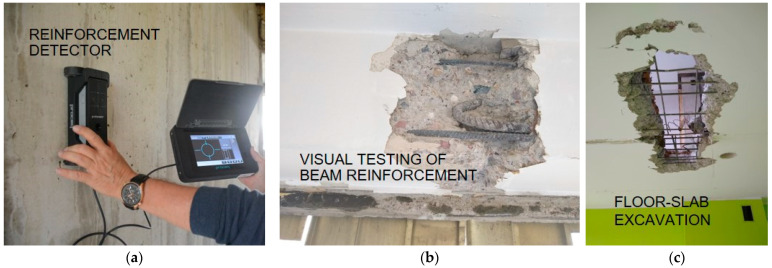
Reinforcement work quality assessment: (**a**) electromagnetic method; (**b**,**c**) excavations.

**Figure 9 materials-17-00018-f009:**
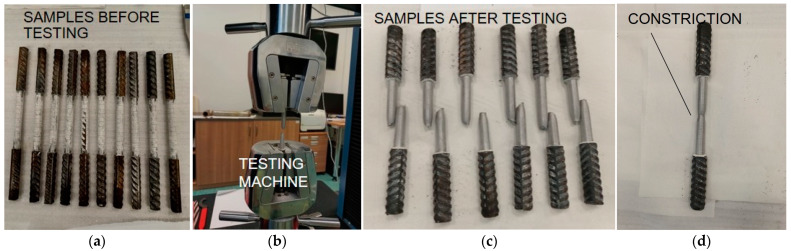
Testing of reinforcing bars: (**a**) rebars before tests; (**b**) testing in the machine; (**c**,**d**) rebars after testing with visible constriction.

**Figure 10 materials-17-00018-f010:**
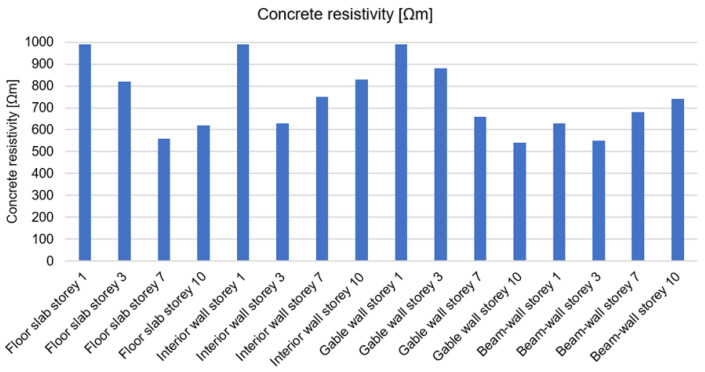
Resistivity of the concrete at the test points read with a Wenner probe.

**Figure 11 materials-17-00018-f011:**
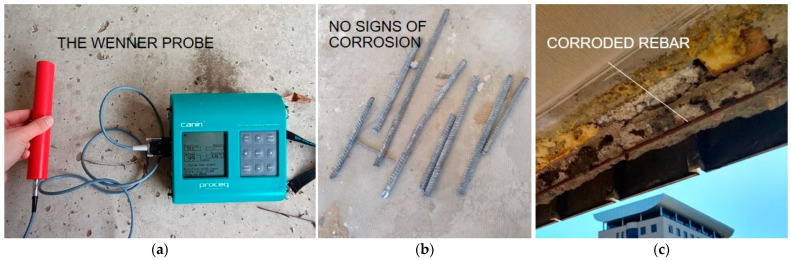
Corrosion investigation of reinforcing steel: (**a**) Wenner probe test; (**b**) visual inspection of rebars from excavations; (**c**) exposed corroded beam-wall rebar.

**Figure 12 materials-17-00018-f012:**
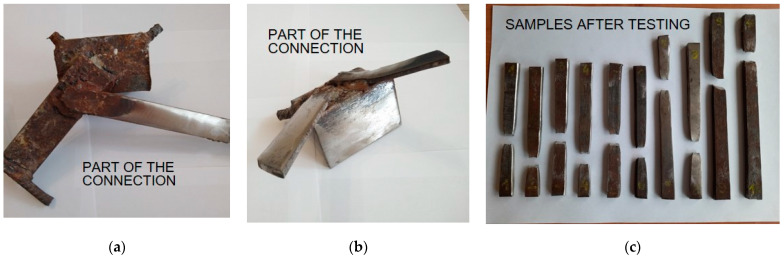
Steel joints (plates) tests: (**a**,**b**) cut-out plates with brushed elements for destructive testing; (**c**) example fragments of torn plates.

**Table 1 materials-17-00018-t001:** Compressive strength results from core drilling and based on empirical relationships obtained.

f_is_ [MPa]	Rebound Median [-]	f_R,ITB_ [MPa]	Δf [%]	f_R,EN_ [MPa]	Δf [%]	f_R,ITB,reduced_ [MPa]	Δf [%]	f_R,EN,scaled_ [MPa]	Δf [%]	f_R,accurate_ [MPa]	Δf [%]
20.10	36.50	28.62	−30%	28.65	−30%	17.17	17%	18.87	7%	20.02	0%
20.83	37.00	29.67	−30%	29.51	−29%	17.80	17%	19.73	6%	20.66	1%
21.00	37.00	29.67	−29%	29.51	−29%	17.80	18%	19.73	6%	20.66	2%
21.17	37.50	30.74	−31%	30.38	−30%	18.45	15%	20.60	3%	21.30	−1%
21.80	38.00	31.83	−32%	31.24	−30%	19.10	14%	21.46	2%	21.94	−1%
22.49	38.50	32.94	−32%	32.11	−30%	19.77	14%	22.33	1%	22.58	0%
23.50	39.00	34.08	−31%	32.97	−29%	20.45	15%	23.19	1%	23.21	1%
23.60	39.00	34.08	−31%	32.97	−28%	20.45	15%	23.19	2%	23.21	2%
23.80	39.50	35.23	−32%	33.84	−30%	21.14	13%	24.06	−1%	23.85	0%
24.50	40.00	36.40	−33%	34.70	−29%	21.84	12%	24.92	−2%	24.49	0%
25.20	40.50	37.59	−33%	35.57	−29%	22.56	12%	25.79	−2%	25.13	0%
25.93	41.00	38.81	−33%	36.43	−29%	23.28	11%	26.65	−3%	25.77	1%

**Table 2 materials-17-00018-t002:** Compressive strength results from the core drillings and based on the empirical relationships obtained from the different structural elements located on storeys 1, 3, 7, and 10 (using the sclerometric method).

Element	Median Number of Rebounds [-]	f_is_ [MPa]	f_R,accurate_ [MPa]	Δf [%]	f_R,reduced_ [MPa]	Δf [%]
Interior wall storey 7	38.5	23.56	22.58	4%	19.77	19%
Gable wall storey 10	39	23.81	23.21	3%	20.45	16%
Floor slab storey 3	39	23.79	23.21	2%	20.45	16%
Gable wall storey 7	39.5	25.71	23.85	8%	21.14	22%
Interior wall storey 1	40	24.23	24.49	−1%	21.84	11%
Floor slab storey 7	40	27	24.49	10%	21.84	24%
Interior wall storey 3	40.5	25.11	25.13	0%	22.56	11%
Floor slab storey 10	40.5	23.51	25.13	−6%	22.56	4%
Gable wall storey 3	41	24.61	25.77	−4%	23.28	6%
Floor slab storey 1	41.5	25.87	26.40	−2%	24.02	8%
Gable wall storey 1	42	23.7	27.04	−12%	24.78	−4%
Interior wall storey 10	42.5	26.11	27.68	−6%	25.54	2%

**Table 3 materials-17-00018-t003:** Compressive strength results from core drilling and longitudinal wave velocities at the surveyed measurement points.

Wave Velocity [km/s]	f_is_ [MPa]
3.572	20.10
3.630	20.83
3.679	21.00
3.709	21.17
3.739	21.80
3.785	22.49
3.815	23.50
3.845	23.60
3.891	23.80
3.922	24.50
3.967	25.20
3.981	25.93

**Table 4 materials-17-00018-t004:** Compressive strength results from the core drilling and from the empirical relationships obtained from the various structural elements located on storeys 1, 3, 7, and 10 (using the ultrasonic method).

Vp [km/s]	f_V,accurate_ [MPa]	f_is_ [MPa]	Δf [%]
3.721	21.66	23.56	9%
3.788	22.55	23.79	6%
3.833	23.20	24.23	4%
3.872	23.80	23.7	0%
3.885	24.01	25.11	5%
3.891	24.11	23.51	−2%
3.898	24.23	23.81	−2%
3.899	24.24	24.61	2%
3.952	25.15	25.71	2%
3.964	25.37	26.11	3%
3.991	25.86	25.87	0%
3.998	25.99	27	4%

**Table 5 materials-17-00018-t005:** Assessment of concrete quality based on longitudinal wave velocity and homogeneity based on reflection number.

Element No.	Average P-Wave Velocity [m/s]	Concrete Quality According to [60]	Median Rebound Number	Calculated Coefficient of Variation of Concrete Compressive Strength *v_f_*	Concrete Homogeneity According to [56]
Wall 1	3816	Good	39	9%	Very good
Wall 2	3726	Good	40	11%	Good
Wall 3	3988	Good	38.5	14%	Medium
Wall 4	3765	Good	38	11%	Good
Wall 5	3811	Good	39.5	12%	Good

**Table 6 materials-17-00018-t006:** Results of rebar diameters.

Identified Steel Grade	Ideal Diameter of Bars with Ribs [mm] [61,62]	Actual Diameter with Ribs [mm]	Design Diameter [mm]	Diameter Obtained Using the Electromagnetic Method [mm]
St3SX	8 (no ribs)	8.62	Φ8	8
		8.67	Φ8	8
		8.68	Φ8	8
		8.87	Φ8	9
		8.79	Φ8	9
18G2	10.3	11.33	Φ10	11
		11.11	Φ10	10
		11.63	Φ10	12
		11.22	Φ10	10
		11.15	Φ10	10
34GS	12.6	13.79	Φ12	14
		13.32	Φ12	12
		13.37	Φ12	12
		13.60	Φ12	14
		12.58	Φ12	12
34GS	14.6	15.00	Φ14	14
		14.88	Φ14	14
		14.98	Φ14	15
		15.04	Φ14	15
		14.73	Φ14	14

**Table 7 materials-17-00018-t007:** Results of the rebar strength tests.

Design Diameter [mm]	Identified Steel Grade	Average Measured Yield Strength f_y_ [MPa]	Minimum Yield Strength f_y,min_ [MPa] [63]	Δf_y_ [%]	Average Measured Tensile Strength f_u_ [MPa]
Φ8	St3SX	395.68	240.00	65%	475.01
Φ10	18G2	458.52	360.00	27%	660.93
Φ12	34GS	475.77	420.00	13%	713.16
Φ14	34GS	470.83	420.00	12%	693.02

**Table 8 materials-17-00018-t008:** Tensile strength test results of connection plates.

Sample Number	Strength Test Determined by Destructive Method	Strength Test Determined by Hardness Test	Δf [%]
	f_u_._is_ [MPa]	f_u_ [MPa]	
1	351	355	−1.1%
2	362	369	−1.9%
3	372	394	−5.6%
4	375	371	1.1%
5	376	399	−5.8%
6	379	360	5.3%
7	400	382	4.7%
8	409	413	−1.0%
9	409	402	1.7%
10	421	424	−0.7%
11	422	424	−0.5%
12	425	427	−0.5%
13	430	442	−2.7%
14	449	462	−2.8%
15	455	482	−5.6%

## Data Availability

The data presented in this study are available upon request from the corresponding author.

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
