# Peer review of "Diagnostics of Large-Panel Buildings—An Attempt to Reduce the Number of Destructive Tests"

_materials, 2023, doi:10.3390/ma17010018_

Round 1
Reviewer 1 Report
Comments and Suggestions for Authors
This paper showed some scientific feature. However, several comments should be addressed.
1. The innovation of this paper should be further presented in the introduction.
2. Insights into non-destructive testing of structures after repair are crucial in the introduction. It is recommended to discuss this within the overview, starting with an exploration of structural characteristics resulting from the repair materials, such as sulfate-aluminum cement, and methods (curing method) employed. Additionally, based on material properties, further extensions can be made to propose modifications. The following references may be helpful to you.
DOI: 10.1016/j.matpr.2021.06.414; 10.1016/j.compositesa.2019.105570
3. In the tables, some symbols lack clear annotations, for example, Δ%
4. For estimating the compressive strength of concrete using the ultrasonic: A regression equation is not sufficient, please provide more discussion on the underlying mechanisms.
5. While the fourth section, 'Discussion,' is commonly presented as an independent section in some articles, its effectiveness as a standalone part is not favorable for this paper. It is suggested to integrate it with the 'Results' section.
Comments on the Quality of English LanguageNo
Reviewer 2 Report
Comments and Suggestions for Authors
The paper investigates an interesting topic such as the diagnostics of large-panel buildings - an attempt to reduce the number of destructive tests. The methdology is pertinent and English is also good. However, some issues need to be considered.
Introduction
The novelties of the paper need to be described in order to support the originality against the existing literature.
The authors considered that the diagnose of the actual conditions of the building is needed in order to avoid undesirable phenomena associated with the failure of a structure, or to identify ways of strengthening. This is generally true. However it is not scientifically described. The authors should discuss the performance of the building and the different ways to stenghten them. They can refer to:
Desai Amit R, Gajjar RK. 2012. Structural control system for mid-rise building. Int J Adv Eng Technol III, II: 30–33. E-ISSN 0976-3945.
Forcellini D, Kalfas K.N. Inter-story seismic isolation for high-rise buildings. Eng Struct 2023;275(2023):115175.
The authors need to demonstrate that "Large-panel structures were erected in many countries, primarily in Central Europe" with the due literature citations.
Section 2
This part is very long and not organized properly. The authors could separate into different subsections the various methods.
Also, are the presented results novel or taken from elsewhere? Please clarify and discuss.
This sentence: "The obtained results indicate that in a facility not located by the sea and not in the zone of a large industrial district, the probability of chloride corrosion is assessed as very low." seems obvuous, please discuss.
Conclusion
This part is more similar to the previous one (discussion). Please reorganize with this information: limitations, applications, future work.
Round 2
Reviewer 1 Report
Comments and Suggestions for Authors
I have checked the revised manuscript, and it can be accepted in current form.
Reviewer 2 Report
Comments and Suggestions for Authors
The paper is now ready for acceptance